

# A Weather Type classification based on the CESM-LE over the Middle Americas.

Yoel A. Cala-Pérez[1], Carlos A. Ochoa-Moya[2], Arturo I. Quintanar[2], and Christopher L. Castro[3]

[1]Posgrado en Ciencias de la Tierra, UNAM. Mexico City, Mexico.
[2]Instituto de Ciencias de la Atmósfera y Cambio Climático, UNAM. Mexico City, Mexico.
[3]Department of Hydrology and Atmospheric Sciences, University of Arizona. Arizona, USA.

**Correspondence:** Carlos A. Ochoa-Moya (carlos.ochoa@atmosfera.unam.mx)

**Abstract.** In this study, two classifications of 20 Weather Types (WTs) were used to identify large-scale and synoptic-scale patterns over the Middle Americas region (MAR) that comprises Mexico, intra-American seas, Central America, and northern South America. The Self-Organizing Maps (SOM) method was used to detect both classifications using standardized Mean Sea-Level Pressure (MSLP) anomalies from the ERA-Interim (ERA-I) reanalysis and the Community Earth System Model-Large Ensemble (CESM-LE) in the historical period and its future projection under an RCP8.5 scenario. The WTs obtained with the CESM-LE in the historical period were assigned to each day of the future projection. Averages of the days belonging to each WT of the historical period were compared between both classifications (ERA-I and CESM-LE) employing seasonal and monthly frequencies of occurrence, correspondence in days of occurrence, Pearson's spatial correlation, and position changes of high-pressure semi-permanent centers such as the North Atlantic Subtropical High (NASH) and the North Pacific High (NPH). From precipitation and MSLP, it was observed that WTs obtained with CESM-LE showed a marked seasonality in their temporal distribution, mainly in the wet period (May-October), similar to the ERA-I classification. Three characteristic phenomena of MAR were the North American Monsoon System (NAMS), the Mid-Summer Drought (MSD), and the Caribbean Low-Level Jet (CLLJ). The CESM-LE adequately represented these phenomena in the historical period compared with ERA-I. Regarding the future projection, the CESM-LE ensemble showed that the spatial patterns were very similar in the historical period. However, differences in precipitation between August and September decreased. To assess the effect of internal climate variability of the CESM-LE, we analyzed the spatial average of precipitation in two regions: NAMS and MSD, for the 34 members of the ensemble in the future projection. In the CLLJ region, differences between the historical and the future projection in terms of averages of the zonal-wind component and precipitation were less than 1 %. This analysis showed that the SOM technique detected the signal of climate change on a regional scale without being affected by the internal global variability of the model. Therefore, SOM emerged as a useful tool for the analysis of numerical experiments such as CESM-LE.

## 1 Introduction

Weather Types (WTs) have successfully been used to analyze possible relationships between synoptic and large-scale atmospheric phenomena at the climate time scales. Therefore it has been the topic of investigation in several studies focused on



how WTs change in frequency and seasonality in future projections and the historical period. Lee and Sheridan (2012) studied projections of future climate scenarios with different $CO_2$ emission rates for the eastern United States (US), combining the Community Climate System Model 3 (CCSM3) outputs with a synoptic WTs classification based on the temperature at 850 hPa from the National Centers for Environmental Prediction/National Center for Atmospheric Research (NCEP/NCAR) reanalyses. Lee and Sheridan (2012) concluded that most of WTs changes in frequency, intensity, and seasonality occurred during periods of carbon doubling in their projections of future scenarios. In contrast, the historical period did not show comparable changes.

Lamb (1972) developed a method to detect a synoptic classification for Europe that comprises 27 WTs. Using MSLP from ERA-Interim and NCEP/NCAR reanalysis, Otero et al. (2018) detected 11 WTs using Lamb (1972) classification and compared to several projections from a selection of Coupled Model Intercomparison Project Phase 5 (CMIP5) models under an RCP8.5 climate scenario, that assumes emissions will continue to increase at the current rate (Stocker et al., 2014). Otero et al. (2018) found that, in the historical period, WTs showed spatial and temporal coherence between the outputs of the models and the reanalyses. However, the frequency of occurrence of the WTs showed changes in their seasonality and geographical distribution in the future period scenario. Two specific WTs identified as "Cyclonic" and "Anticyclonic" circulations showed notable differences in frequency. Otero et al. (2018) used them to estimate precipitation and temperature behavior changes in the European region. Demuzere et al. (2009) found that the Fifth Generation General Atmospheric Circulation Model (ECHAM5) coupled with the Max Planck Institute - Ocean Model (ECHAM5-MPI/OM) could reproduce those WTs for the October-April period detected by Lamb (1972). Despite the inter-annual variability observed between RCP4.5 and RCP8.5 future climate scenarios, the trend was similar for each WT. The ECHAM5-MPI/OM model showed an increase in the frequency of WTs with anticyclonic circulation and a decrease in the patterns with cyclonic circulation and easterlies (Demuzere et al., 2009), a result similar to Otero et al. (2018).

Climatological studies that used synoptic classifications to analyze WTs behavior, such as changes in the frequency of occurrence and dynamics, appeared almost at the same time climate models began to be used (Sheridan, 2002). Using model outputs in the WTs approach is based on different comparison criteria, such as analysis of correlations, persistence, and frequency of the WTs detected principally from reanalyses data. For example, Belleflamme et al. (2013) studied the behavior of WTs obtained with geopotential height values at 500 hPa over Greenland using ERA40 reanalysis data and compared the detected WTs in other reanalyses such as NCEP/NCAR and Twentieth Century Reanalysis version 2 (20CRv2), as well as with the representation of WTs in model outputs of the CMIP3 and CMIP5 for RCP4.5 and RCP8.5 scenarios. The authors used the Spearman correlation coefficient and the Euclidean distance to compare WTs obtained from model outputs and reanalyses, employing geopotential height anomalies using WTs mean values from ERA40. Belleflamme et al. (2013) detected almost all WTs from ERA40 in CMIP3 and CMIP5 model outputs. However, the observed frequency and persistence of the WTs in ERA40 were not those of the historical period. For future projections, CMIP3 and CMIP5 model outputs showed different frequencies and persistence for the same WTs.

Furthermore, Gibson et al. (2016) used 24 CMIP5 models to reproduce WTs obtained with Mean Sea Level Pressure (MSLP) values from 20CRv2 in Australia. They used three criteria to evaluate those models most capable of representing the historical period: frequency, persistence, and transitions among WTs. The models that best reproduced the WTs in terms of MSLP





overestimated frequency and persistence, while those least able to replicate them did not show a clear trend in frequency and persistence.

Different variables and methods have been used to develop synoptic classifications based on WTs. After testing different numbers of WTs and using MSLP from the 20CRv2 reanalysis, Gibson et al. (2016) generated a climatology of 30 WTs over Australia employing the SOM method and detected them in three other reanalyses and 24 CMIP5 models in the historical

period. The differences in spatial resolution determined the ability to reproduce the WTs among models. Models with high resolution had a better performance reproducing the WTs from the 20CRv2 (Gibson et al., 2016).

Some studies focused on the Continental United States (CONUS), for example, Prein et al. (2019) and Prein et al. (2016). Prein et al. (2016) used MSLP, precipitable water, and wind speed at 700 hPa from ERA-Interim reanalysis to detect 12 WTs combining three methods: Principal Component Analysis, hierarchical cluster analysis, and k-means. They analyzed precipi-

tation, separating trends associated with changes in precipitation intensity and changes in the frequency of WTs. Prein et al. (2016) found that changes in precipitation intensity led to an increasing precipitation trend over the Midwest and Northeast. In contrast, for the Southwest, WTs changes in frequency were linked to decreased precipitation. Also, based on ERA-Interim data, Prein et al. (2019) tested different WTs numbers and input variables to come up with a 12-WT classification similar to that found by Prein et al. (2016). Prein et al. (2019) assigned the 12 WTs to daily model outputs with two different domain

extents, one corresponding to the North American domain in the Coordinated Regional Climate Downscaling Experiment (NA-CORDEX) and the second to the Weather Research and Forecasting model with a grid-space of 36 km (WRF36) and 24 ensemble members. The authors found that data in the NA-CORDEX domain better reproduced the WTs than WRF36. Furthermore, Prein et al. (2019) observed an increase in the frequency of WTs representing the North American Monsoon system (NAM), indicating a greater monsoon flow over the southwestern CONUS and northwestern Mexico.

Other studies have focused on obtaining WTs associated with the behavior of precipitation in the region of Mexico, the intra-American seas, and northern South America, using the SOM method to analyze precipitation separating WTs in wet and dry regimes (Díaz-Esteban and Raga, 2018). Using the k-means method, Sáenz and Durán-Quesada (2015) studied the behavior of low-level wind regimes over Central America, and Moron et al. (2016) analyzed moisture transport in the Caribbean. These studies generated WT classifications utilizing reanalysis data, such as ERA-Interim (Díaz-Esteban and Raga, 2018; Sáenz and

Durán-Quesada, 2015) and NCEP (Moron et al., 2016). Díaz-Esteban et al. (2020) employed CMIP5 model outputs during the historical period to generate a WT classification to compare against the corresponding WTs from the ERA-Interim reanalysis. The authors found that the evaluated CMIP5 models adequately represented the observed WTs behavior in ERA-Interim Díaz-Esteban and Raga (2018).

Over the region of Mexico, the intra-American seas, and northern South America, Ochoa-Moya et al. (2020) analyzed 20

WTs that represent large-scale climatological circulation patterns containing the behavior of the main climate features. With this synoptic classification based on WTs detected using standardized MSLP anomalies from the ERA-Interim reanalysis data, Ochoa-Moya et al. (2020) showed that the position of large-scale pressure centers such as the North Pacific High (NPH) and the North Atlantic Subtropical High (NASH) influence significantly the behavior of phenomena related to precipitation regimes in Mexico and Central America (Ochoa-Moya et al., 2020; Cook and Vizy, 2010) such as the NAMS (Adams and Comrie,





1997; Vera et al., 2006), the Caribbean Low-Level Jet (CLLJ) (Wang, 2007; Cook and Vizy, 2010; Muñoz et al., 2008) and the Mid-Summer Drought (MSD) over southeastern Mexico and Central America (Vera et al., 2006; Magaña et al., 1999).

To our knowledge, no studies explore the behavior of WTs in future climate projections to understand the synoptic and large-scale circulation changes over the tropical region of Mexico, the intra-American seas, and northern South America. This study aims to analyze how the WT classification identified by Ochoa-Moya et al. (2020) will behave in the future in terms of their 100 frequency and representation of phenomena that influence precipitation in Mexico and Central America. Section 2 presents a brief description of the data and the methodology used, while Section 3 contains the results. Conclusions and recommendations for future research projects are in Section 4.

## 2 Data and Methods

We used the SOM method as a pattern classification technique. The SOM method was an artificial intelligence algorithm based 105 on an unsupervised artificial neural network (Kohonen, 2001). This technique was commonly used in meteorology to help reduce the dimensionality of data (Ochoa-Moya et al., 2020). We used the trained SOM neural network to obtain a set of WTs that resulted from minimizing the euclidean distance to a centroid. The node from the SOM network mostly "rewarded" in the search becomes the node representing the WT (Hewitson and Crane, 2002; Gibson et al., 2016). Following Ochoa-Moya et al. (2020) methodology, we detected 20 WTs in the CESM-LE data over the MAR region.

The study region used to obtain the WTs classification corresponded to 0-50° N and 10-150° W (Figure 1). The area defined this way was apt to identify large-scale and synoptic phenomena such as the NASH and the NPH relevant to weather and climate variability over Mexico, the inter-American seas, Central America, and northern South America. We referred to it as the Middle Americas Region (MAR) in Ochoa-Moya et al. (2020) (see Figure 1 red box). Over the MAR region, the SOM method detected the main modes of precipitation variability associated with regional phenomena such as NAMS, MSD, and 115 the CLLJ during the wet period from May to October (Ochoa-Moya et al., 2020; Jáuregui, 1971).

We followed a two-pronged approach to gain insight into the long-term changes that WTs have undergone and the significance that future projections under a climate change scenario have for the MAR region. Firstly, we built a WTs historical baseline using the WTs classification of Ochoa-Moya et al. (2020) obtained from the ERA-Interim reanalysis (ERA-I) 1980-2016 period for the MAR region. Secondly, we applied the SOM method to the Community Earth System Model-Large Ensemble 120 (CESM-LE) output to obtain a new classification of WTs. The CESM-LE project aimed at a climate change evaluation using 40 ensemble members covering the recent past and near future with a spatial resolution of $1° \times 1°$ for the 1920-2100 period and a pre-industrial control period extending from year 400 until 1850 based on the radiative forcing baseline of 1850. The 40 ensemble members originated from slightly perturbed initial conditions and accounted for the historically observed and future internal atmospheric variability (Kay et al., 2015). Due to some systematic differences in the solutions and data availability, we em-125 ployed only 34 ensemble members (https://www.cesm.ucar.edu/projects/community-projects/LENS/known-issues.html). We constructed daily standardized MSLP anomalies based on the climatological mean of 1980-2005, weighed by latitude $\sqrt{cos\theta}$.





These data were fed to the SOM (Kohonen, 2001) neural network code with both ERA-I reanalysis data and CESM-LE output. We thus obtained two sets of WTs, which we denoted WT-R and WT-M, for the ERA-I and CESM-LE, respectively.

As mentioned earlier, previous studies (for instance, (Belleflamme et al., 2013; Gibson et al., 2016; Otero et al., 2018)) have
dealt with the problem of comparing WTs from reanalysis and future climate scenarios. In this study, we opted to use Pearson's spatial correlation and changes in the position of pressure centers (euclidean distance in degrees) of the two main large-scale circulation patterns affecting the daily weather in the MAR region, namely, the NASH and the NPH. Thus, we first compared WT-R and WT-M in terms of the criteria mentioned above for the historical period. Interpolation from the CESM-LE to ERA-I grid size ($0.7° \times 0.7°$) was performed by employing bi-linear interpolation to apply Pearson's correlation method.

We used Precipitation data from Climate Hazards InfraRed Precipitation (CHIRP) to evaluate changes in precipitation associated with each WT from the WT-R classification. CHIRP was a database based on satellite precipitation estimates that combined Thermal InfraRed precipitation estimates with Cold Cloud Duration rainfall estimates with a fixed threshold of 235 K and horizontal resolution of $0.25°$, covering the globe between $50°$ N and $50°$ S (Funk et al., 2015).

The evaluation of the WT-M from CESM-LE RCP8.5 projected future scenario involved assigning the WT-M classification
to daily MSLP data in the 2006-2100 period. To this end, we searched for the smallest Euclidean distance between MSLP standardized anomalies from each projected day and the averaged MSLP standardized anomaly of each 20 WT-M classification set. The WT assignment in future projection was applied to the ensemble mean and the 34 ensemble members.

A bootstrapping technique was used to statistically test differences in the frequency of occurrence of WTs among the classifications mentioned above using 10,000 permutations at a 90 % confidence level. We employed the same technique to test both
comparison criteria: Pearson's spatial correlation and the position changes of the NASH and NPH.

## 3 Results

### 3.1 Relationship between the WTs obtained from CESM-LE and the WTs from ERA-Interim.

Figure 2 showed the behavior of the frequency by month of each WT-R for the 1980-2016 timespan and WT-M for the 1920-2005 model historical period (HP) and the WT-M for the future projection period (FP) 2006-2099. Applying Jáuregui (1971)
seasonal precipitation classification during the HP, both WT classifications 1 through 10 presented a clear-cut monthly distribution for the wet period (May-October). In contrast, the next WTs, 11 through 20, showed a widespread monthly distribution for the dry-cold (November-February) and dry-warm (March-April) periods. Further analysis of the monthly frequency of occurrence (see Figure 2) revealed some salient differences between WT-R and WT-M classifications. These were manifest during the dry-warm period and the last months of the wet period (September-October). Since MSLP was the variable used by SOM
to construct the WTs then, it was natural to associate WTs monthly frequency of occurrence with the amplitude and pattern differences between the reanalysis and the model in MSLP.

The subsequent analysis focused on the WTs that influence the behavior of precipitation in the region of Mexico and Central America (see Figure 1), which were prevalent during the wet period (WT-R1 to WT-R10 and WT-M1 to WT-M10). Past studies showed that during the precipitation annual cycle, a peak was present over Northwest Mexico and a double peak over southern





Mexico and Central America. These features represented two modes of precipitation variability in terms of water availability in the region (Adams and Comrie, 1997; Magaña et al., 1999).

  Analyzing the number of coincidental days revealed similarities between WTs classifications in HP. Figure 3 showed the number of days in the period 1980-2005 in which each WT-R coincided with each WT-M. We called this index Correspondence In Days (CID). Comparisons between WT-R1 to WT-R10 and WT-M1 through WT-M10 showed higher CID values than those

165 in the dry period, which confirmed the observed seasonality. Figure 3 showed that WT-R1 and WT-M5 concurred for 161 days during 1980-2005 (around 27 % of the total number of days when WT-R1 was present). September was a month when both WT-R1 and WT-M5 presented their highest frequency of occurrence, 47 % and 48 %, respectively (see Figure 2).

  Determining how close the synoptic configuration was between WT-R1 and WT-M5 played a salient role in defining whether both WTs represented the same atmospheric features. Calculations of comparison criteria mentioned above were presented in

the upper panels of Figure 4 for MSLP and Precipitation corresponding to WTs in the wet period. A comparison between WT-R1 and WT-M5 resulted in significant correlation values for MSLP and Precipitation of 0.86 and 0.72, respectively (see Figure 4). These correlation values could be related to a similar frequency of occurrence during September (see Figure 2) and a CID value around 27 % (see Figure 3).

  Euclidean distance calculations for the NASH and the NPH between ERAI and CESM-LE data for 1980-2005 during the

175 wet period (WTs 1 through 10) were presented in bottom panels from Figure 4. In order to compare WTs classifications, pairs of WTs were selected from the model and reanalysis based on the criteria mentioned earlier, which led us to conclude that these pairs of WTs had similar synoptic atmospheric circulation patterns as well (see Table 1). Comparisons of WT-R1 and WT-M5 exhibited that their differences in euclidean distances were not statistically significant for both the NPH and the NASH, presenting relatively small values in this metric compared with other WTs in the wet period (see Figure 4). In other words,

both WTs could correctly represent the locations of the NPH and the NASH centers with significant spatial correlations, which implied that they contained similar synoptic configuration characteristics.

## 3.2 Synoptic analysis

Figure 5 showed MSLP and temperature at 2 meters (T2m) anomalies for WTs 1 through 10. All WTs in the model and reanalysis showed positive temperature anomalies over the continental land masses north of about 30° N. Guided by the

185 frequency diagrams of reanalysis and model data, we found the following two WT pairs for the onset of the NAM season: 1) WT-R9 and WT-M8 (May) and 2) WT-R4 and WT-M2 (June). Inspection of the first pair of WTs in Figure 5 indicated positive temperature anomalies along the Sierra Madre Occidental (SMO) of about 1.5° C with somewhat larger positive anomalies in the reanalysis compared to the model. Temperature anomalies for the second pair of WTs revealed a similar pattern as in the first pair of WTs but with more prominent temperature anomalies of about 2-3° C along the SMO.

Further analysis revealed another set of pairs, WT-R5 and WT-M1, representing the mature stage of the NAM season during August. Here, negative temperature anomalies along the southern portion of the SMO in reanalysis data were found. However, the model showed negative anomalies in Southern Mexico and Central America. Another WT pair of interest was found for WT-R1 and WT-M5, representing the decaying stage of the monsoon season in September. At this point in the monsoon



evolution, WT-R1 shows negative temperature anomalies of less than 1° C along the SMO. In contrast, WT-M5 showed positive
temperature anomalies. This feature indicated a weaker monsoon development in the CESM ensemble output compared to
reanalysis data. Of note were the semi-permanent atmospheric circulation centers such as NASH and NPH, which showed
a well-defined configuration over the North Atlantic and North Pacific, respectively. In general, the WTs-M showed steeper
horizontal gradients in the isobaric structure of these systems, as evidenced in the WT pairs mentioned above.

Analyzing MSLP anomalies in Figure 6 revealed that most of the WTs from 1 to 10 in both classifications (WTs-R and WTs-
M) presented positive MSLP anomalies in the region where NASH and NPH were located. Near Central America, the Gulf of
Mexico, and Mexico, negative MSLP anomalies were present, implying low-pressure systems typical of summer convection.

The WTs-M also showed similar anomalous precipitation patterns compared to the WTs-R (see Figure 6). WT-R1 and WT-
M5 showed that CESM-LE (see WT-M5) underestimated the precipitation behavior in the study region compared to the data
from CHIRP (see WT-R1). Hence, anomalously positive precipitation values prevailed over the continents in most WTs, while
negative values were observed over the eastern Pacific and western Atlantic ocean basins (see Figure 6). It is important to note
that both WTs classifications detected the northward shift of the ITCZ in the summer months as indicated by the dipole pattern
north of 5° N (see Figure 6).

To complete the above discussion on how the WTs classification aided in interpreting the NAM evolution in the region,
we turned to Figure 6, which showed precipitation and MSLP anomalies. To that end, we looked at May's first pair of WTs,
namely, WT-R9 and WT-M8. Inspection of this pair revealed a negative precipitation anomaly over southeastern Mexico and
the Central America corridor. Additionally, there was an apparent negative precipitation anomaly along the SMO typical of
the onset of the NAM. The pair WT-R4 and WT-M2 (June) showed an increase in positive precipitation anomaly over the
SMO, which was indicative of the beginning of the NAM. In the reanalysis, this pair also showed increased precipitation over
southern Mexico and Central America, whereas the model showed weaker positive precipitation anomalies.

Inspection of WT-R5 and WT-M1 corresponding to August showed a clear mature monsoonal pattern of positive precipita-
tion anomalies along the SMO. WT-R5 showed a positive precipitation anomaly as far north as 30° N, while WT-M1 precip-
itation anomaly could only reach up to 25° N. Analysis of the September pair, WT-R1, and WT-M5, revealed a precipitation
anomaly pattern retracting towards the south along the SMO.

### 3.3 Detection of important climate features over Mexico and Central America.

The climate behavior of precipitation in Mexico responds fundamentally to NAMS and MSD. In southeastern Mexico and
northern Central America, the observed behavior of the MSD precipitation exhibited a maximum in September and a minimum
in August (Magaña et al., 1999). On the other hand, in the southwestern US and northwestern Mexico, where the NAMS
was defined, precipitation increased in July-August while presenting a decrease in September-October (Adams and Comrie,
1997; Vera et al., 2006). In this respect, Figure 2 showed that the frequency of occurrence for WT-R1 and WT-M5 peaked
in September, while WT-R5 and WT-M1 peaked in August. These two pairs of WTs were present during the NAMS and
the characteristic MSD minimum. Ochoa-Moya et al. (2020) detected these two phenomena comparing mean precipitation of
WT-R5 and WT-R1 (see Figure 7a in Ochoa-Moya et al. (2020)). In Figure 7, we performed the same procedure with WT-M1





and WT-M5, which were spatially correlated with WT-R5 and WT-R1, respectively (see Table 1 and Figure 4 for confidence levels).

Figure 7 showed over the NAMS region (see Figure 1 purple box) a positive precipitation difference, which means that the accumulated precipitation decreased from August to September. However, in the MSD region (see Figure 1 blue box), a negative precipitation difference indicated an increase from August to September. This result evidenced the ability of the SOM technique to detect the spatial pattern of precipitation associated with these two climate phenomena in the model and reanalysis data.

Another phenomenon that influenced the behavior of weather and climate conditions in MAR, with relevance to Mexico and Central America (Mo et al., 2005), was the CLLJ. Wang (2007) found that CLLJ intensity had an annual distribution with minima in May and October and maxima in January and July and was related to position changes of the NASH high-pressure center. Wang (2007) used winds at 925 hPa to detect the CLLJ signal. CESM-LE did not have the 925 hPa level available. Instead, we detected this phenomenon using zonal wind data at 850 hPa with the CESM-LE ensemble outputs.

Figure 8 presented the CLLJ as the averaged fields of the most frequent WTs during the months associated with maxima and minima for zonal wind intensity (Wang, 2007). Mean zonal wind values (U) in the region defined by Wang (2007) showed maxima during January and July (WT-M17 and WT-M1, respectively) and minima during May and October (WT-M8 and WT-M10, respectively). Following the wind direction, the CLLJ influenced the contribution of moisture to the Yucatan Peninsula and the Gulf of Mexico during May and July (WT-M8 and WT-M1, respectively), which was notable in WT-M1 due to an

intensification of the CLLJ. In the case of January and October (WT-M17 and WT-M10, respectively), the CLLJ blew westward to Central America. However, at this point in the season, the contribution of moisture flux was considered not to be important (Wang, 2007). However, WT-10 had higher precipitation anomalies over the Yucatan Peninsula, which could be related to the influence of convection and ocean heating from the summer months.

    Next, in Figure 8, we examined the seasonal evolution of the CLLJ and its relation to the high-pressure center of the NASH.
The synoptic configuration that the NASH featured over the Atlantic Ocean corresponded to the WT-M1 and WT-M8 patterns. A result in agreement with the findings of Wang (2007) and Ochoa-Moya et al. (2020). Indeed, in the presence of WT-M1 and WT-M8, the NASH high-pressure center was closer to the CLLJ region (43° W and 40° W, respectively), that is, slightly displaced to the west, which resulted in a CLLJ intensification. WT-M17 and WT-M10, on the other hand, showed NASH's position further east of the CLLJ region (29° W and 34° W, respectively). This difference in the mean position of the NASH

could be related to a CLLJ northward excursion, as evidenced in the WT-M1 and WT-M8 MSLP patterns, indicating a possible moisture transport to eastern Mexico and southern US (see Figure 8).

    Mean zonal wind values showed similar magnitudes to those reported by Wang (2007) and Ochoa-Moya et al. (2020) in the CLLJ box indicated in Figure 8 (see Table 2). This analysis showed that the model and the reanalysis were in close agreement, highlighting the SOM method's ability to detect the CLLJ as a predominant phenomenon in MAR.



## 3.4 WTs frequency changes in the CESM-LE future projection.

Up to this point, we have analyzed the frequency of occurrence of WTs (see Figure 2) and the ability of CESM-LE in the historical period to reproduce important climatic features compared to ERA-I. This study aimed to analyze WTs changes in their frequency of occurrence under a climate change scenario and explore the CESM-LE internal variability from a WT climatology perspective. To this end, we started showing the WTs frequencies relative to the total number of days (expressed as a percentage) for the wet period. Figure 2) featured relative frequencies for WTs. To analyze changes, we compared the first-row corresponding to the HP with the remaining rows corresponding to the 34 ensemble members of the FP.

WT-M1 shows the highest frequency in both periods (HP and FP), while WT-M9 had the lowest frequency of occurrence. From Figure 2, we learned that WT-M1 peaked in frequency during July-August (wet period). On the other hand, WT-M9 showed low frequencies of occurrence along the annual cycle, with a small peak in November corresponding to the dry-cold period (see Figure 2). Note that differences in relative frequencies among ensemble members showed small but significant differences (see Figure 9), which could be related to the internal climate variability of the model (Kay et al., 2015).

Analyzing changes in WT relative frequencies of occurrence (from WT-M1 to WT-M10) in the FP, concerning the behavior of their relative frequencies of occurrence in the HP (see Figure 9 right). WeT observed consistency between each member of CESM-LE within the same WT. Some WTs, such as WT-M1, WT-M3, WT-M6, and WT-M9 show a significant increase in relative frequency in most of their members. Other WTs, such as WT-M2, WT-M4, WT-M5, WT-M7, and WT-M10 show a significant decrease in their frequencies in most of their members. It is noteworthy that WT-M1 and WT-M4 present the greatest changes in the frequency of occurrence, showing an increase and a decrease, respectively. As we have previously analyzed, WT-M5 and WT-M1 are important to understand the behavior of the NAMS and MSD phenomena due to their higher frequency in September and August, respectively (see Figure 2). WT-M5 shows a significant decrease in its frequency of occurrence during FP, while WT-M1 shows a significant increase.

## 3.5 WTs precipitation averages and their changes in the future projection.

Figure 10 shows the same difference as in Figure 7, but for the ensemble average in the FP. Note that in Figure 10, positive (negative) values indicated more (less) precipitation during August (September), which corresponded to the mature NAMS (August) and the second maximum in MSD (September). Additionally, we observed a similar spatial distribution of precipitation compared to HP. Positive precipitation differences appeared in Northwestern Mexico, while negative precipitation differences were apparent over the Yucatán peninsula. Focusing on NAMS and MSD regions (inset in Figure 1), the averaged precipitation anomaly during the HP resulted in 0.54 $mmday^{-1}$ and 0.28 $mmday^{-1}$ for the FP, suggesting a decrease in precipitation over the NAMS region (inset in Figure 1). On the other hand, over the MSD region (inset in Figure 1), a decrease from -0.81 $mmday^{-1}$ in the HP to -1.1 $mmday^{-1}$ we interpreted as a strengthening in the magnitude of the MSD phenomenon.

To further analyze precipitation changes for the WT-M classification from every ensemble member, we spatially averaged precipitation over NAMS and MSD regions (inset in Figure 1). We computed the precipitation average only for the days of the month where each WT-M had its highest frequency of occurrence. Accordingly, Figure 11 contained differences in precipitation





average between each ensemble member in the FP and the average in the HP. Comparing plots in Figure 11, it was evident that changes in the MSD region indicated a decrease in precipitation for almost all ensemble members. In the case of the NAMS, the signal was entirely different because only WT-M1 and WT-M4 featured a decrease in precipitation across all ensemble members, while the rest showed mixed signals.

As mentioned above, WTs detected in HP were assigned to the FP. We observed that WT-M1 and WT-M5 were still the main WTs preserved in the FP (see Figure 9) and contributed the most to the precipitation in August and September, respectively (see Figure 11 left column). The precipitation differences in WT-M1 associated with a mature NAMS and a minimum in MSD showed a decrease in precipitation compared to HP. By contrast, WT-M5 presented differences close to zero for the NAMS region and precipitation decrease over the MSD region. Wind intensity and precipitation over the CLLJ region did not show noticeable differences (around 1 %) in the FP when contrasted with the HP.

## 4   Conclusions

This study obtained two weather type (WTs) classifications using reanalysis data (ERA-I) and CESM-LE model outputs in the historical period (HP). We classified WTs employing the SOM method to determine synoptic and large-scale variability in the MAR region. Regarding the frequency of occurrence, the CESM-LE performed better during the wet period (May-October) than the dry period (November-April). When comparing reanalysis and model outputs, most notably, the annual cycle temporal distribution of WTs coincided in both classifications. Utilizing criteria such as CID, we established relationships among WTs in both classifications. We found that WT-R5 and WT-R1 were close in spatial distribution to WT-M1 and WT-M5, respectively.

Selected WTs from 1 to 10 in both classifications (Reanalysis and CESM-LE model) showed a high frequency of occurrence during the wet period. Accordingly, precipitation and MSLP significant spatial correlations were higher than 0.7 between both classifications. The WTs also showed a marked seasonality in the temporal distribution of the future projection (FP), conserving their frequency behavior in the same months as in the HP. However, WTs such as WT-M1 (August) and WT-M5 (September) showed a change in the frequency of occurrence in FP compared to HP, with an increase and decrease, respectively.

The SOM method showed the ability to recognize patterns of seasonal precipitation variability in the presence of phenomena such as NAMS, MSD, and CLLJ, both in the HP and in the FP of the CESM-LE model. Both periods (HP and FP) presented a decrease in precipitation from August to September in the NAMS region and an increase in the MSD region. These results coincided with those reported in the literature. However, even though NAMS and MSD featured a similar behavior for the FP, the most frequent WTs during these months (WT-M1 and WT-M5, respectively) showed changes in their frequency of occurrence and the accumulated precipitation averages. We performed this analysis for 34 ensemble members. We found statistically significant differences among them compared to the HP, which may be related to the model's internal climate variability.

The CLLJ behaved similarly for the HP and the FP, with minor changes in the spatial distribution of wind and precipitation fields (around 1 % for the FP). This result suggested a possible relationship between the CLLJ and non-significant changes in the NASH high-pressure center position.



The presence of climate variability is an issue that has prevented the adequate evaluation of long-term trends in temperature and precipitation in climate change scenarios. The SOM technique is immune to this problem as it was based on pattern recognition detection through a neural network application. A technique that did not rely on linearized statistics from a covariance error matrix, such as the EOF technique. Precipitation showed significant differences in the outputs of the CESM-LE ensemble

models. However, separation by WTs served as a filter for the model's internal climate variability. The SOM technique was able to recognize the behavior of the main phenomena associated with precipitation variability in the MAR region by detecting WTs with only MSLP standardized anomaly as input. In this way, it was possible to observe a well-defined seasonal behavior of the WTs and their accumulated precipitation averages in the wet, dry-cold, and dry-warm periods.

The relevance of this study hinges on evaluating and processing climate model outputs from a perspective of WTs classifi-

cation and its changes in future climate projections. The SOM technique proved a powerful tool for analyzing and assessing multiple model outputs. A climatic catalog developed by Ochoa-Moya et al. (2020), informed us about distinct synoptic conditions that we detected and analyzed in a climate projection. Finally, regional studies on precipitation utilizing Large Ensembles datasets help assess the uncertainties among global climate models and gain insight into precipitation changes over the MAR region.

*Author contributions.* YACP and CAOM planned the study; YACP and CAOM analyzed the data; YACP, CAOM and AIQ wrote the manuscript draft; CAOM, AIQ, and CLC reviewed and edited the manuscript.

*Competing interests.* The authors declare that they have no conflict of interest.

*Acknowledgements.* This study was partially funded by project PAPIIT IA103916 and IA102916. YAC was supported by Mexican National Council for Science and Technology (CONACyT) scholarship 722415. We also acknowledge the CESM-LE Community Project. All figures

were generated with The NCAR Command Language (Version 6.6.2) [Software]. (2019). Boulder, Colorado: UCAR/NCAR/CISL/TDD. http://dx.doi.org/10.5065/D6WD3XH5



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



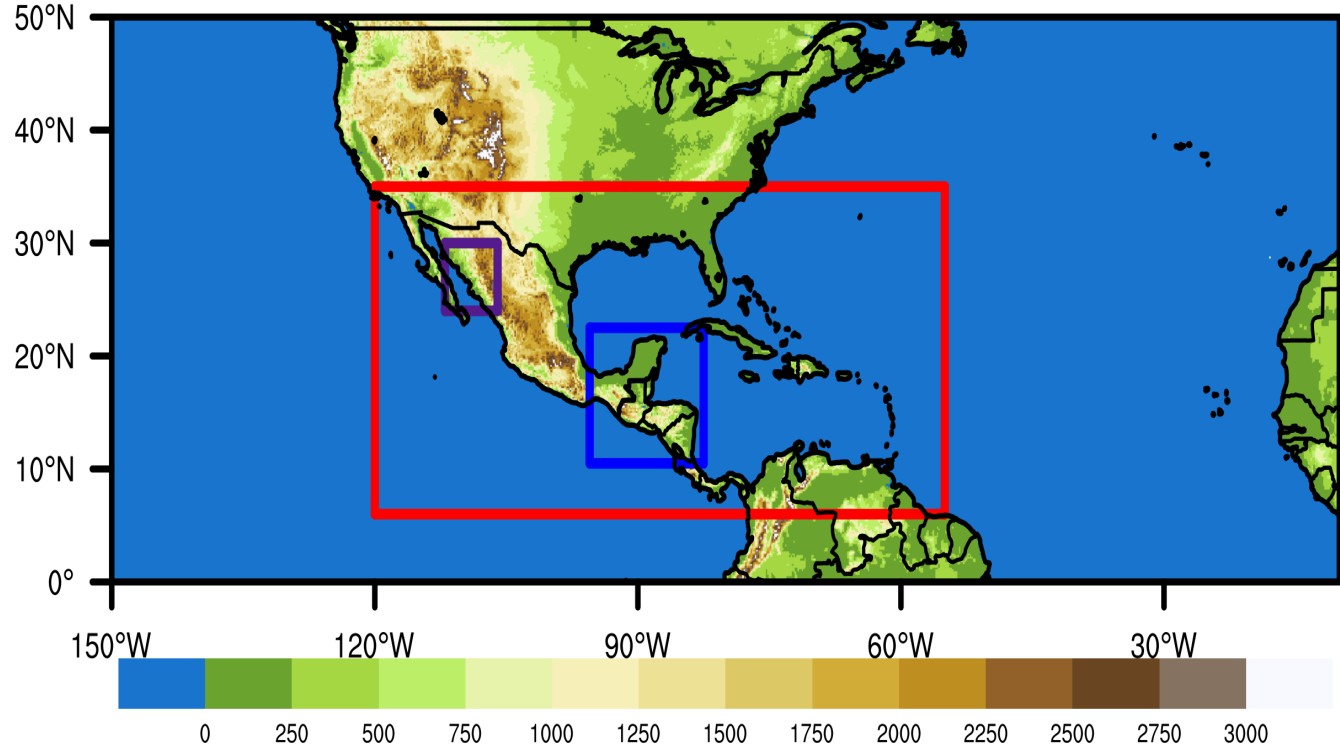

**Figure 1.** *Study region (0-50° N and 10-150° W). Red box: approximate MAR domain (5-35° N and 55-120° W). Purple box: approximate NAMS domain (24-30° N and 106-112° W). Blue box: approximate MSD domain (10.5-22.5° N and 82.5-95.5° W)*



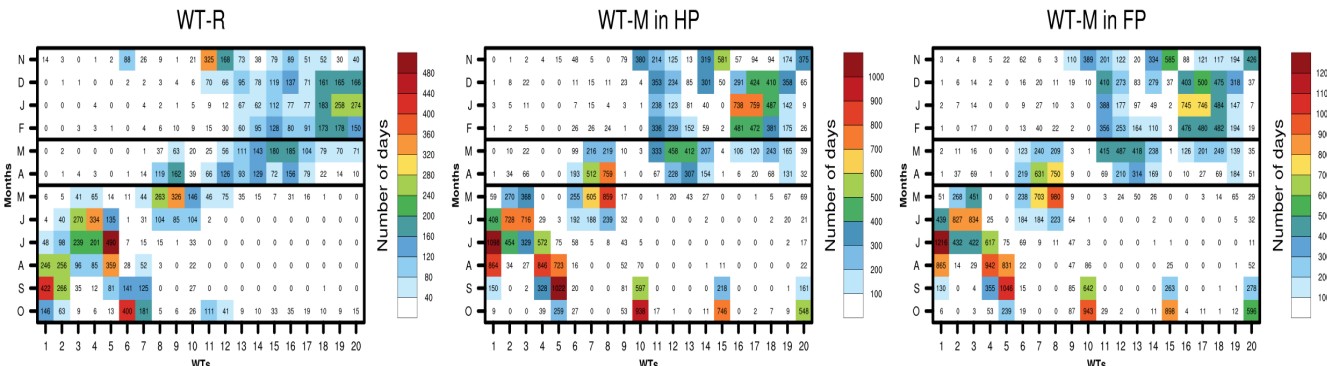

**Figure 2.** *Monthly frequency of occurrence (in days) for each WT-R (left), each WT-M for CESM-LE ensemble in HP (center) and each WT-M for CESM-LE ensemble in FP (right). Horizontal thick lines divide the annual march into 3 periods: a dry-cold period (November–February); a dry-warm period (March–April); and a wet period (May–October) from Jáuregui (1971).*



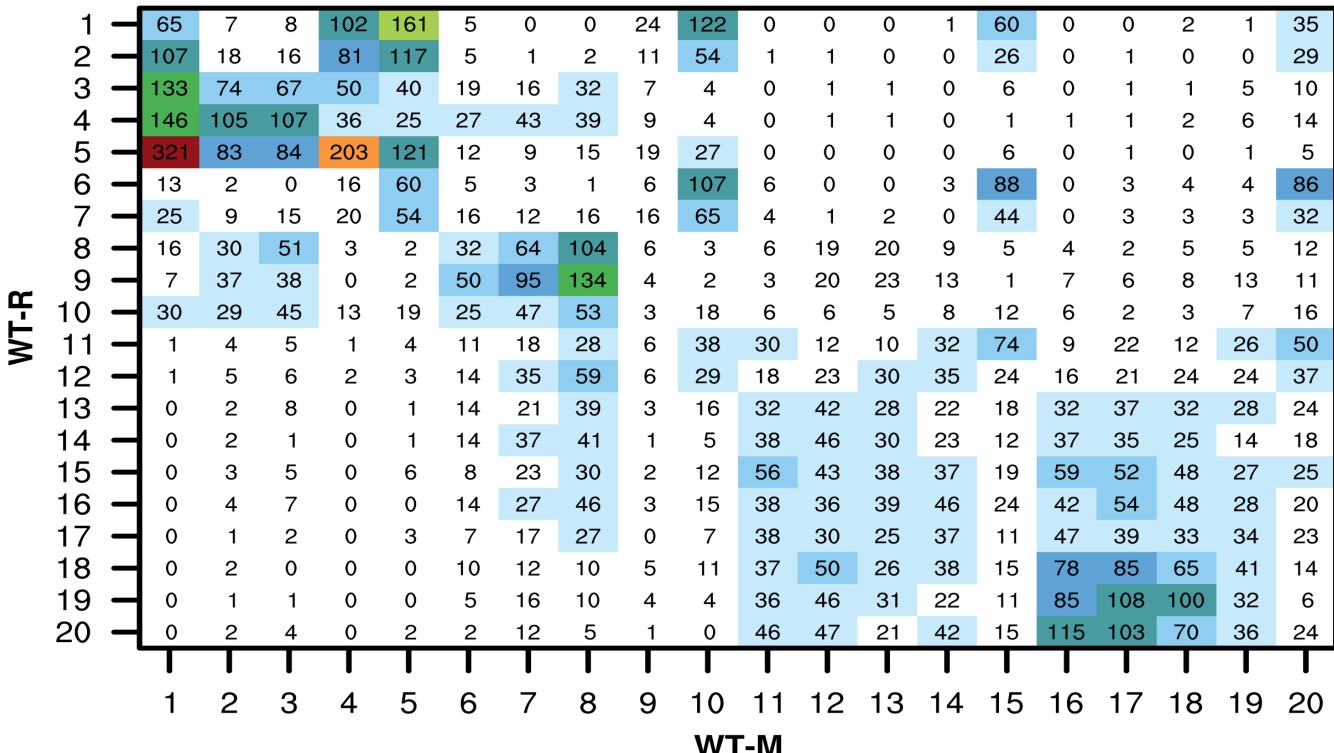

**Figure 3.** *Correspondence in days (CID) for the period 1980-2005 between WT-R and WT-M.*





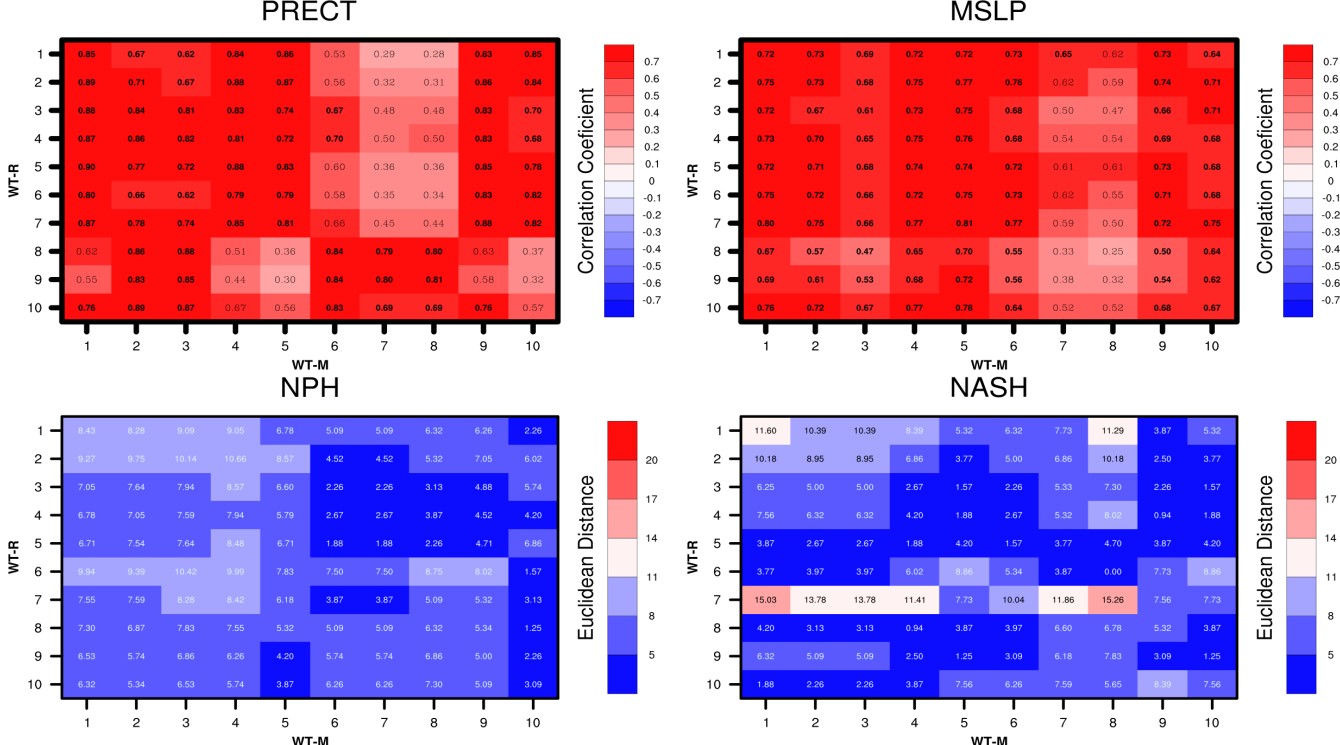

**Figure 4.** *Top row: Pearson spatial correlation between WT-R and WT-M classifications for precipitation (left) and MSLP (right). Bottom row: the NPH (left) and the NASH (right) euclidean distance (in degrees) of their high-pressure center position between WT-R and WT-M classifications. Both comparison criteria were based on the period 1980-2005. Bold values indicated statistical confidence at the 90 % level.*



**Figure 5.** Composite averages of MSLP fields (contours from 988 to 1028, every 2 hPa) and T2m (colors) for WTs R1-R10 (two upper rows) and M1-M10 (two bottom rows) for the period 1980-2005.



**Figure 6.** Composite averages of MSLP anomaly fields (-16 to 13 contours, every 1 hPa) and precipitation anomaly fields from CHIRP (colors) for WT-R1 to WT-R10 (two upper rows) in the period 1980-2005. WT-M1 to WT-M10 (two bottom rows) similar to WT-R classification but with CESM-LE data. Solid (dashed) lines represent positive (negative) anomalies. The bold contour line is zero.



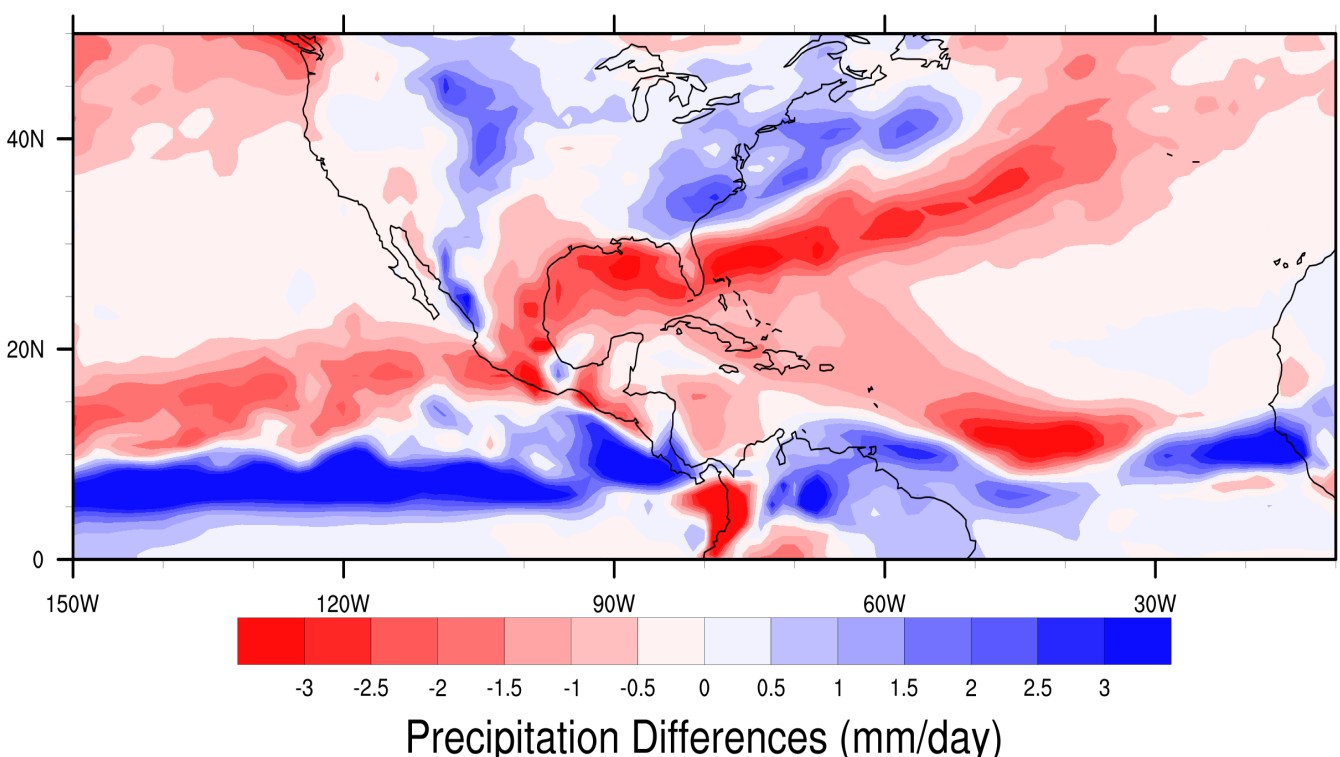

**Figure 7.** *CESM-LE averaged precipitation from WT-M1 minus WT-M5 for the period 1980-2005. WT-M1 precipitation field was obtained from the days of occurrence in August and WT-M5 from September, according to their highest frequency of occurrence by month in Figure 2*



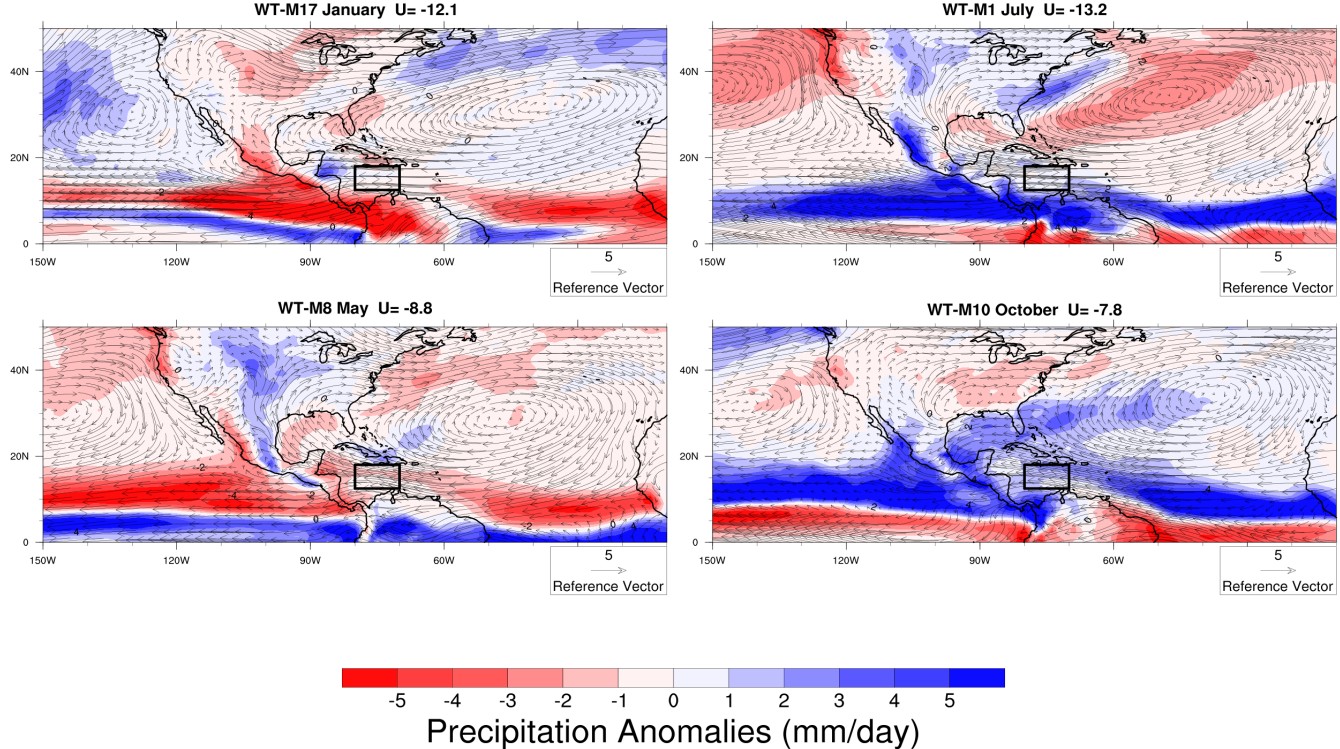

**Figure 8.** *Composite wind averages at 850 mb (arrows in $ms^{-1}$) and anomaly precipitation fields (colors). Above each plot was indicated the WT and the month for the composite. The box bounded area (lower left corner at 12.5° N and 80° W; upper right corner at 18° N and 70° W) corresponds to the Caribbean Low-Level Jet (CLLJ) core defined by Wang (2007).*





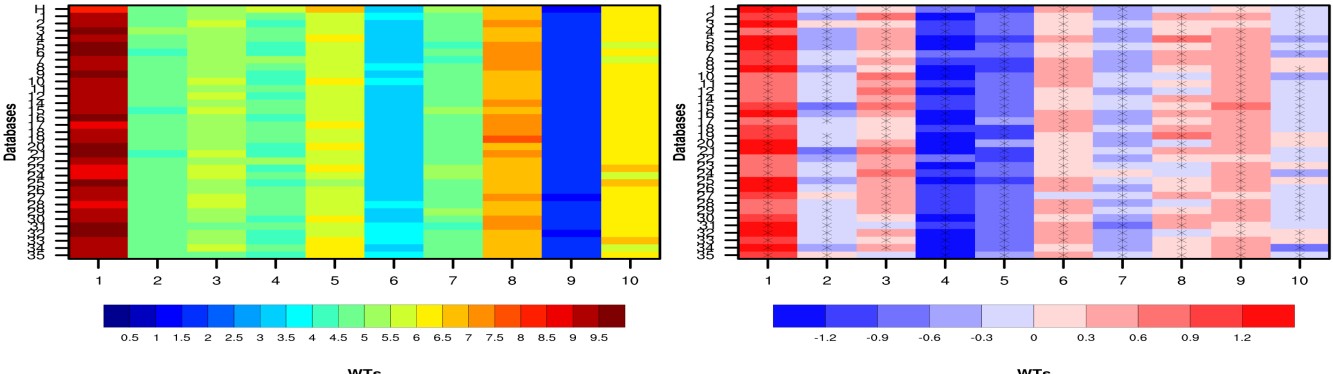

**Figure 9.** *Left: Normalized frequencies of occurrence (%) for M1-M10 in the historical period 1920-2005 (upper row) and for the 34 members of CESM-LE in the future projection 2006-2100 (remaining rows). Right: Difference in the behavior of relative frequencies of occurrence (%) for M1-M10 in the 34 members of CESM-LE for the period 2006-2100 compared to the historical period 1920-2005. Relative frequency of occurrence is obtained by dividing by the total number of days. Markers represent significant differences at 90 % confidence.*





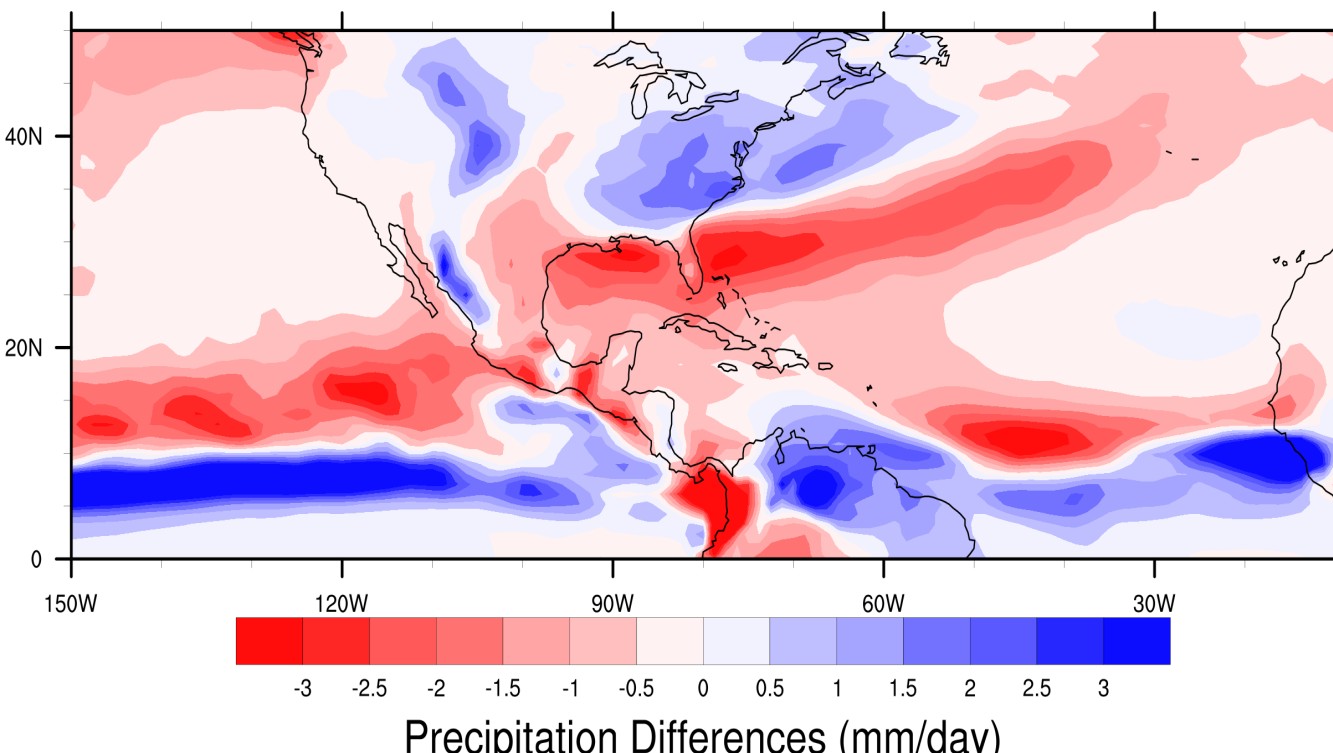

**Figure 10.** *Same as in Figure 7, but for the period 2006-2100*





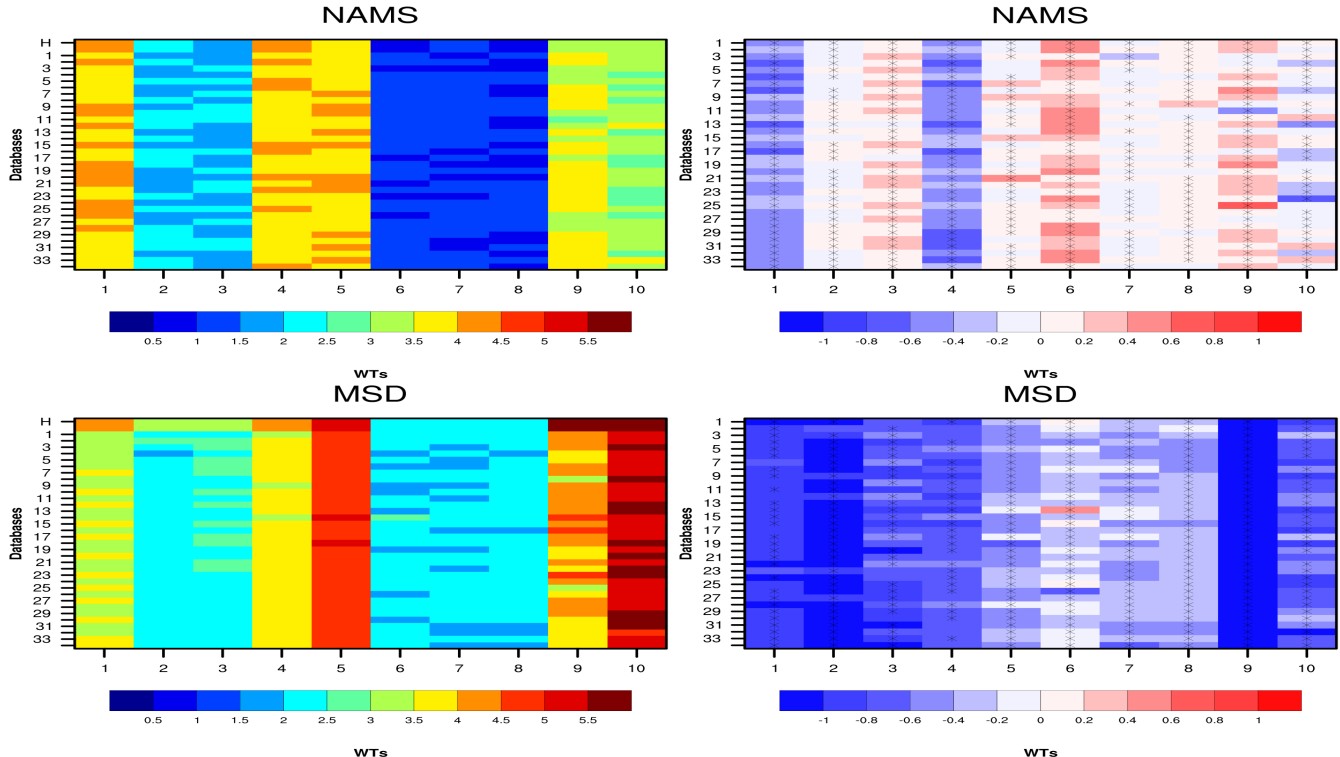

**Figure 11.** *Precipitation behavior for the NAMS and MSD domains. Left: Behavior of accumulated average precipitation (mm/day) for M1-M10 in the historical period 1980-2005 (upper row) and for the 34 members of CESM-LE in the future projection 2006-2100 (remaining rows). Right: Difference in the behavior of the accumulated average precipitation (mm/day) for M1-M10 in the 34 members of CESM-LE for the period 2006-2100 with respect to the historical period 1980-2005. Markers represent significant differences at 90 % confidence.*





| WT-R | WT-M | Month | CID | Corr_MSLP | Corr_PREC | ED_NPH | ED_NASH |
|------|------|-------|-----|-----------|-----------|--------|---------|
| 1 | 5 | Sep | 161 | 0.72 | 0.86 | 6.78 | 5.32 |
| 5 | 1 | Jul | 321 | 0.72 | 0.90 | 6.71 | 3.87 |
| 6 | 10 | Oct | 107 | 0.68 | 0.82 | 1.57 | 8.86 |
| 9 | 8 | May | 134 | 0.32 | 0.81 | 6.86 | 7.83 |

**Table 1.** Summary of the comparison criteria used to relate the WTs-R with the WTs-M. "Corr" means correlation and "ED" means Euclidean distance (in degrees).





| Month (WT-R/WT-M) | U-Wang | Pcp-Wang | U-Ochoa | Pcp-Ochoa | U-Model | Pcp-Model |
| :---: | :---: | :---: | :---: | :---: | :---: | :---: |
| # | $m\ s^{-1}$ | $mm\ day^{-1}$ | $m\ s^{-1}$ | $mm\ day^{-1}$ | $m\ s^{-1}$ | $mm\ day^{-1}$ |
| January (R20/M17) | -10.5 | 0.7 | -9.0 | 0.9 | -12.1 | 1.92 |
| July (R5/M1) | -12.0 | 2.4 | -11.6 | 1.94 | -13.2 | 2.55 |
| May (R9/M8) | -9.0 | 2.4 | -9.7 | 1.7 | -8.8 | 1.35 |
| October (R6/M10) | -6.5 | 4.5 | -6.5 | 4.28 | -7.8 | 3.47 |

**Table 2.** Zonal wind (U) and Precipitation (Pcp) area average for the maxima and minima reported by Wang (2007) and calculated with SOM Catalog for both WTs classifications: WTs-R ( indicated as Ochoa-Moya et al. (2020)) and WTs-M (indicated as Model)