# Peer review of "A Weather Type classification based on the CESM-LE over the Middle Americas."

_EGUsphere, 2022_

## Referee Comment (RC1)

**Manuscript egusphere-2022-1332 Review**

January 5, 2023

Manuscript # egusphere-2022-1332 entitled "A Weather Type classification based on the CESM-LE over the Middle Americas".

This manuscript analyses weather patterns (WTs) in the Intra-Americas Seas region classified using ERA-Interim data and the CESM-LENS1 through the Self-organising maps (SOMs) technique. The manuscript argues that it demonstrates the usefulness of the SOM technique to detect a climate change signal and separate the different atmospheric states that compose the main features of the climate of the region.

The study is very well-written and has the potential to be a valuable contribution to our understanding of how climate models represent WT patterns and their frequency over the Middle-Americas region and how models project these characteristics to change in the future. However, the manuscript frequently overlooks and disregards relevant scientific literature, the main methodology is not well described, and more importantly, I have several major concerns about the analysis and presentation quality. Given the amount of revisions I think are required to get this manuscript suited for a WCD publication is so large, my recommendation is to reject the manuscript in its present form.

**Major comments**

1. The SOM methodology and the evaluation of present-day climate of the CESM LENS lacks detail and in its current form has several shortcomings. Firstly, the manuscript does not describe the SOM technique in a way that is reproducible or understandable by any reader. Secondly, the authors make no attempt to defend their choice of using 20 WTs. Previous studies (Gibson et al., 2016; Zhao et al., 2020) extensively describe the SOM methodology including an objective determination of the number of WTs/nodes to be used. In its present form, the manuscript uses 20 WTs but several of them look very much alike (W or appear to occur very infrequently (WT-M9). For example, the authors compare WT-R1 and WT-M5 for given their correlation coefficients of 0.86 and 0.72 with precipitation and MSLP, respectively. However, WT-M1 exhibits correlation coefficients of 0.85 and 0.72, respectively. Their correlation coefficients in Figure 4 would suggest that *all* WTs from ERA-Interim are well correlated with three WTs from CESM (WT-M1,4,5). Another example is WT-M9, which is found in less than 0.5% of the days (Fig. 9). Therefore, I am not convinced the WT patterns are significantly different amongst themselves in the same dataset, and this makes me wonder how

relevant are the comparisons of model WTs with ERA-Interim WTs. My recommendation is for the authors to firstly, better explain their methodology and, secondly, to conduct a more thorough analysis of the model and reanalysis WTs, as in Gibson et al. (2016).

2. The authors compute anomalies of 2-m temperature and precipitation without explicitly explaining how these anomalies are computed or their interpretation, which is fundamental to understand the relevance of each WT for known climate features such as the Midsummer drought (MSD). Based on my several readings of the manuscript, these anomalies are differences from the annual mean, which in my view, makes the interpretation of all the precipitation anomaly figures difficult. For example, in Figure 5, I can't distinguish differences in the precipitation patterns between WT-R1, WT-R2 and WT-R5, and the same is true for WT-M1, WT-M4 and WT-M5. Their reference ? computes anomalies as differences from the annual cycle (seasonality removed). My suggestion is to (1) carefully describe how the anomalies are computed, and for the wet season composites (MSD and NAM-related) to use deseasonalized anomalies in order to demonstrate that the SOM is finding patterns associated, e.g., with the NAM onset or the MSD.

3. The authors claim the SOM is able to diagnose the characteristic patterns of the NAM and MSD based on the sign of the anomalies, which as I have said in my previous comment, are difficult to interpret as deviations from the annual mean. Also, the authors make no attempt to compare their patterns (Fig. 7) with previous studies of the NAM (Geil et al., 2013; García-Franco et al., 2021) or the MSD (Zermeño-Díaz, 2019; Zhao et al., 2020; Zhao and Zhang, 2021). I would argue the pattern of the MSD diagnosed by these previous studies is not similar to what is shown in Fig. 7. The authors also claim that their WTs are able to detect the CLLJ variability but I am also not convinced, the manuscript needs to show that WTs are able to capture the intraseasonal variability of the CLLJ and its influence on continental precipitation (see e.g. García-Franco et al., 2022), because otherwise, their evidence in Figure 8 could just be due to the different months chosen for their composites and not necessarily a skillful diagnosis of the CLLJ by the SOM (see e.g. Martinez et al., 2019, for schematics on the seasonality of the CLLJ and the ITCZ precipitation).

4. Section 3.5 is very interesting and potentially a great contribution but the way its presented makes it difficult to understand whether differences between historical period and future period are due to frequency changes or changes in the precipitation associated with each WTs. My suggestion is to better disentangle both factors and provide enough statistical evidence to demonstrate which factor is more relevant. A secondary suggestion would be not to include the 2006-2030 period in the future period composites, and instead use 2030-2100 to better highlight differences between present-day conditions and model projections of future climate. I also think showing ensemble-mean differences between these two periods is sufficient to show changes due to the scenario forcing.

5. Relevant literature is overlooked or ignored several times.

   (a) The first two sentences in the introduction have no references at all. These two sentences must cite the most relevant studies that support the claim that SOMs are a useful tool to diagnose WTs, specially for future projections.

   (b) Several, many of them recent, studies have diagnosed patterns associated with the MSD and NAM which are ignored in the manuscript, in particular the study by Zhao et al. (2020) who used SOMs to diagnose the characteristic patterns of the MSD.

     - Barlow, M., Nigam, S., and Berbery, E. H. (1998). Evolution of the north american monsoon system. *Journal of Climate*, 11(9):2238–2257
     - Zermeño-Díaz, D. M. (2019). The spatial pattern of midsummer drought as a possible mechanistic response to lower-tropospheric easterlies over the intra-americas seas. *Journal of Climate*, 32(24):8687–8700
     - Zhao, Z., Holbrook, N. J., Oliver, E. C., Ballestero, D., and Vargas-Hernandez, J. M. (2020). Characteristic atmospheric states during mid-summer droughts over Central America and Mexico. *Climate Dynamics*, 55(3)
     - García-Franco, J. L., Chadwick, R., Gray, L., Osprey, S., and Adams, D. K. (2022). Revisiting mechanisms of the mesoamerican midsummer drought. *Climate Dynamics*, pages 1–21

   (c) Studies on the impact of the NASH on the CLLJ and the MSD are ignored, particularly those that would make claims made in the manuscript contentious, e.g., that the NASH impacts the MSD directly (e.g. Herrera et al., 2015). More recent references to links between the CLLJ and the NASH are also needed.

     - Herrera, E., Magaña, V., and Caetano, E. (2015). Air–sea interactions and dynamical processes associated with the midsummer drought. *International Journal of Climatology*, 35(7):1569–1578
     - Martinez, C., Goddard, L., Kushnir, Y., and Ting, M. (2019). Seasonal climatology and dynamical mechanisms of rainfall in the caribbean. *Climate dynamics*, 53(1-2):825–846
     - García-Martínez, I. M. and Bollasina, M. A. (2020). Sub-monthly evolution of the caribbean low-level jet and its relationship with regional precipitation and atmospheric circulation. *Climate Dynamics*, 54(9):4423–4440
     - Zhao, Z., Han, M., Yang, K., and Holbrook, N. J. (2022). Signatures of midsummer droughts over Central America and Mexico. . *Climate Dynamics*. doi:https://doi.org/10.1007/s00382-022-06505-9

**Minor comments**

- (l 96) The MSD does not occure solely on southeastern Mexico but also in southwestern and eastern Mexico (Perdigón-Morales et al., 2018; Zhao and Zhang, 2021).

- Figure 2. Frequency is shown in units of total days, which makes it difficult to compare the three panels shown here as all of them have different colorbars due to different sample size. My suggestion is to use days/month as a unit, computed individually for each dataset/panel.

- The writing is for the most part clear and easy to follow. However, the manuscript tends to write sentences in the past tense that ought to be in the present tense. For example, the description of the NAMS [l 220-224] or the influence of the CLLJ [l. 235] are written in the past tense which is incorrect. The method section [l 104-106] has the same issues.

- (l 143) The bootstrapping technique to determine statistically significant differences needs to be better explained. Is this sampling with or without replacement? How are the distributions grouped and how are they compared?

- (l 163) The correspondence in days (CID) index needs to also be better explained. Are the comparisons done for specific days, say, January 8th, 2000? or what is correspondence, exactly? Since the CESM-LENS are not initialized forecasts, I am not sure what is the relevance of finding coincidental WTs on the same exact date.

- In Section 3.2, the authors relate WTs to various stages of the NAM and make statements about the strength of the NAM in the CESM-LENS. However, I am not convinced that the authors provide enough evidence for these statements. For instance, they claim the onset of the NAM is associated with positive temperature anomalies and negative precipitation anomalies, but this seems counter intuitive. Deseasonalized anomalies or at least, a qualitative comparison with previously published patterns for each stage (Barlow et al., 1998; Geil et al., 2013; García-Franco et al., 2021) is required.

- (l. 243-245) Since the authors are inferring moisture transport in this sentence, it would be valuable to note if this relationship between CLLJ and moisture transport agree with previous studies (Martinez et al., 2019; Perdigón-Morales et al., 2021), given a strong CLLJ?

- For Figures 9 and 11 I would suggest to compute differences in the ensemble-mean only, perhaps as probabilities.

- (l 318) "results coincided with those reported in the literature." What literature exactly? Please compare with, at least, the references I've suggested.

- (l 324) "This result suggested a possible relationship between the CLLJ and non-significant changes in 325 the NASH high-pressure center position." Not sure if I understood this sentence but I don't think there is enough evidence to support it. Rewrite and expand what you mean.

**Technical corrections**

l273 "WcT" is this a typo?

**References**

Barlow, M., Nigam, S., and Berbery, E. H. (1998). Evolution of the north american monsoon system. *Journal of Climate*, 11(9):2238–2257.

García-Franco, J. L., Chadwick, R., Gray, L., Osprey, S., and Adams, D. K. (2022). Revisiting mechanisms of the mesoamerican midsummer drought. *Climate Dynamics*, pages 1–21.

García-Martínez, I. M. and Bollasina, M. A. (2020). Sub-monthly evolution of the caribbean low-level jet and its relationship with regional precipitation and atmospheric circulation. *Climate Dynamics*, 54(9):4423–4440.

García-Franco, J. L., Osprey, S., and Gray, L. J. (2021). A wavelet transform method to determine monsoon onset and retreat from precipitation time-series. *International Journal of Climatology*, 41(11):5295–5317.

Geil, K. L., Serra, Y. L., and Zeng, X. (2013). Assessment of cmip5 model simulations of the north american monsoon system. *Journal of Climate*, 26(22):8787–8801.

Gibson, P. B., Uotila, P., Perkins-Kirkpatrick, S. E., Alexander, L. V., and Pitman, A. J. (2016). Evaluating synoptic systems in the cmip5 climate models over the australian region. *Climate Dynamics*, 47(7):2235–2251.

Herrera, E., Magaña, V., and Caetano, E. (2015). Air–sea interactions and dynamical processes associated with the midsummer drought. *International Journal of Climatology*, 35(7):1569–1578.

Martinez, C., Goddard, L., Kushnir, Y., and Ting, M. (2019). Seasonal climatology and dynamical mechanisms of rainfall in the caribbean. *Climate dynamics*, 53(1-2):825–846.

Perdigón-Morales, J., Romero-Centeno, R., Ordóñez, P., and Barrett, B. S. (2018). The midsummer drought in Mexico: perspectives on duration and intensity from the CHIRPS precipitation database. *International Journal of Climatology*, 38:2174–2186.

Perdigón-Morales, J., Romero-Centeno, R., Ordoñez, P., Nieto, R., Gimeno, L., and Barrett, B. S. (2021). Influence of the madden-julian oscillation on moisture transport by the caribbean low level jet during the midsummer drought in mexico. *Atmospheric Research*, 248:105243.

Zermeño-Díaz, D. M. (2019). The spatial pattern of midsummer drought as a possible mechanistic response to lower-tropospheric easterlies over the intra-americas seas. *Journal of Climate*, 32(24):8687–8700.

Zhao, Z., Han, M., Yang, K., and Holbrook, N. J. (2022). Signatures of midsummer droughts over Central America and Mexico. . *Climate Dynamics*. doi:https://doi.org/10.1007/s00382-022-06505-9.

Zhao, Z., Holbrook, N. J., Oliver, E. C., Ballestero, D., and Vargas-Hernandez, J. M. (2020). Characteristic atmospheric states during mid-summer droughts over Central America and Mexico. *Climate Dynamics*, 55(3).

Zhao, Z. and Zhang, X. (2021). Evaluation of methods to detect and quantify the bimodal precipitation over Central America and Mexico. *International Journal of Climatology*, 41:E897–E911.